# Study on Mechanism and Verification of Columnar Penetration Grouting of Time-Varying Newtonian Fluids

Xuguang Zhao [1,2,3], Zhiquan Yang [1,2,3,*], Xiangrui Meng [4], Shaobin Wang [4], Rui Li [5], Hanhua Xu [6], Xiangpeng Wang [7], Changwen Ye [7], Tianbing Xiang [8], Wanzhong Xu [1,2,3], Junzhi Chen [1,2,3], Ping Wang [9], Jinke Yuan [7] and Xiaohui Li [7,10]

[1] Faculty of Public Safety and Emergency Management, Kunming University of Science and Technology, Kunming 650093, China
[2] Key Laboratory of Geological Disaster Risk, Prevention and Control and Emergency Disaster Reduction of Ministry of Emergency Management of the People's Republic of China, Kunming 650093, China
[3] Key Laboratory of Early Rapid Identification, Prevention and Control of Geological Diseases in Traffic Corridor of High Intensity Earthquake Mountainous Area of Yunnan Province, Kunming 650093, China
[4] Jinhui Mining Co., Ltd., Longnan 742300, China
[5] Gansu Jinhui Xinke Material Co., Ltd., Longnan 742300, China
[6] Yunnan Key Laboratory of Geotechnical Engineering and Geohazards, Kunming Prospecting Design Institute of China Nonferrous Metals Industry Co., Ltd., Kunming 650051, China
[7] State Key Laboratory of Geohazard Prevention and Geoenvironment Protection, Chengdu University of Technology, Chengdu 610059, China
[8] Power China Kunming Engineering Co., Ltd., Kunming 650051, China
[9] Zhejiang Huadong Construction Engineering Co., Ltd., Hangzhou 310014, China
[10] Chengdu University of Technology W-G-P Engineering Technology Co., Ltd., Chengdu 610059, China
[*] Correspondence: yzq1983816@kust.edu.cn

**Abstract:** Penetration grouting technology is an important technical means to improve the mechanical properties of gravel soil layers, and the time-varying characteristics of Newtonian fluid viscosity have an important influence on the morphology and effect of penetration grouting. However, these time-varying properties are not considered in the current research on the mechanism of Newtonian fluid penetration grouting. In this paper, by studying the basic rheological equation of Newtonian fluids and its dynamic viscosity time-varying law, the penetration motion equation of viscosity time-varying Newtonian fluids is discussed, by means of theoretical analysis and experimental research. Based on this, the time-varying viscosity Newtonian fluid columnar penetration grouting diffusion mechanism (TVNCPGDM) equation is derived, the application scope of the equation is analyzed and a grouting experiment is designed to verify it. The results show that the theoretical value of the grouting diffusion radius calculated by the TVNCPGDM equation, is closer to the experimental value than that obtained by the equation of columnar penetration grouting without considering the viscosity time-varying Newtonian fluid, with a 12.9% improvement in accuracy. This shows that the TVNCPGDM equation derived in this paper, can better reflect the diffusion law and diffusion morphology of column penetration grouting of Newtonian fluid, which changes with time in the injected medium; and the diffusion radius obtained for penetration grouting is more in line with the actual grouting engineering demands. The research results can provide some theoretical guidance for the actual grouting of loose gravel soil layers.

**Keywords:** Newtonian fluid; viscosity degeneration; columnar penetration grouting mechanism; grouting experiments

## 1. Introduction

The gravel soil layer is a kind of loose coarse debris accumulation layer formed in the Quaternary period. At present, many geological disasters (landslides, collapses, debris flows, etc.) are induced in the loose gravel soil layer. For example, Sichuan Wenchuan

Niujuangou (the epicenter of the Wenchuan earthquake), Yunnan Dongchuan Jiangjiagou, and Nujiang Dongyue basins are all loose gravel soil layers. Geological disasters often occur in these areas, cause major casualties and huge property losses, and destroy the ecological environment [1–4]. Engineering practice shows that [5–8] grouting technology has become the preferred effective technical means to solve the current problems of various types of gravel soil development stratum engineering, especially penetration grouting technology.

Penetration grouting is one of the earliest geotechnical reinforcement techniques used in the field of grouting. It was first used by French engineer Charles Berlghy, in the repair of Dieppe's scour gate in 1802 [5]. Because penetration grouting technology causes less disturbance to the gravel layer, it is widely used in civil engineering applications such as railways, highways, mines, and tunnels. Wang et al. [9] applied lateral radiation grouting technology to solve the uneven settlement problem of a highway bridge transition section, in the coastal area of Jiangsu. Qi et al. [10] used the small pipe advance grouting technique to solve the construction problems of a tunnel crossing a weak fault and fractured surrounding rock, in Beijing. Zeng et al. [11] used Ansys finite element software to simulate and analyze the strengthening effect of small duct grouting, which has some practical engineering value. According to the poor mechanical properties of saturated soft loess, Zhang [12] used the splitting grouting method to reinforce a subway station in Xi'an, which greatly improved the soil strength. Zeng et al. [13] used in-tunnel deep hole grouting and ground tracking grouting to reinforce the surrounding rock of the subway tunnel between Youfang Street Station and Binlu Station of Harbin Metro Line 3, to improve the bearing capacity of the soil. In order to avoid the large deformation of the large-span chlorite schist in the Lianchengshan Tunnel from Baoji to Hanzhong, Chen et al. [14] used radial grouting of the surrounding rock for reinforcement, which effectively prevented the occurrence of large deformation disasters in the tunnel. Hou et al. [15] proposed the grouting reinforcement technology of shallow hole sealing and grouting, and deep hole reduction reinforcement of fractured surrounding rock in deep roadways, which solved the common problems of surrounding rock strength deterioration and stress environment deterioration in deep roadways.

The grouting mechanism plays a decisive role in the grouting effect of the gravel soil layer. According to different rheological constitutive equations, grouting fluids can be divided into Newtonian fluids, Bingham fluids, and power-law fluids [6,7]. The penetration grouting of these fluids in the injected medium or material, can show three diffusion morphologies: spherical, columnar, and column-hemispherical [7]. A Newtonian fluid, is one in which the shear stress at any point is a linear function of the shear deformation rate. Non-Newtonian fluids, are fluids that do not obey the Newtonian law of viscosity. Kotha, G. et al. [16,17] studied the properties of steady-state non-Newtonian fluids. At present, many research results have been obtained on the mechanism of Newtonian fluid penetration grouting, such as Maag sphere, sleeve valve, and columnar diffusion theory [6]; Zou [18] analyzed the diffusion law of Newtonian fluids in plane radial circular cracks; Baker [19] studied the calculation equation of the maximum diffusion radius of Newtonian fluids in rock fractures; Liu [20] studied the diffusion radius equation of Newtonian fluids in fracture surfaces; Jiang [21] discussed the columnar diffusion mechanism of Newtonian fluids under horizontal and vertical drilling conditions; Louis [22] derived its diffusion equation in two-dimensional rough fractures; Yang [23] studied the diffusion mechanism of Newtonian fluids in column-hemispherical penetration grouting. Liu et al. [24] used a rotational viscometer to test the rheological properties of ordinary Portland cement slurry under different conditions, and obtained the influence of the water–cement ratio, temperature, and hydration time, on the rheological properties. Wang et al. [25] established a two-stage column-hemispherical diffusion model of Newtonian fluids, based on the intrinsic constitutive equation, penetration equation of motion, and fractal theory, by considering the curvature of porous media, and verified the accuracy of the model through numerical simulation and a series of slurry injection tests, which improves the accuracy of the calculation compared with the traditional model, and has good guidance significance

for engineering practice. Fu et al. [26] developed a grout penetration model for three fluids, by considering the dead weight of the slurry, and the calculation results were in agreement with the actual test results.

However, the current results of these Newtonian fluid penetration grouting mechanisms do not consider their time-varying characteristics. When calculating the theoretical diffusion size, it is considered that the viscosity is constant throughout the grouting process, and the calculated theoretical diffusion size is much larger than the measured value in the actual grouting. Therefore, the research on the mechanism of columnar penetration grouting of viscosity time-varying Newtonian fluids in this paper, can provide some theoretical support for practical grouting construction projects such as railways, highways, mines, and tunnels, and through the TVNCPGDM equation, a more accurate penetration grouting diffusion radius can be calculated, to meet the needs of practical engineering.

## 2. Rheological Equation of Time-Varying Newtonian Fluid

The basic rheological equation of a Newtonian fluid is as follows [1]:

$$\tau = \eta\gamma \tag{1}$$

According to the time-varying law of dynamic viscosity of Newtonian fluids in reference [27]:

$$\eta(t) = \eta_0 e^{kt} \tag{2}$$

Combining Equations (1) and (2), the rheological equation of a viscosity time-varying Newtonian fluid is:

$$\tau = \eta(t)\gamma = \eta_0 e^{kt}\gamma \tag{3}$$

where $\tau$ is shear stress; $\gamma$ is the shear rate ($\gamma = -\frac{dv}{dr}$); $\eta$ is dynamic viscosity; $\eta_0$ is the initial value of the dynamic viscosity of the Newtonian fluid, which is approximately equal to the fixed dynamic viscosity $n$ in Equation (1). $\eta(t)$ is the dynamic viscosity of the Newtonian fluid at time $t$; $t$ is the grouting time; and $k$ is a time-varying coefficient, which can be measured experimentally.

## 3. Study on Columnar Penetration Grouting Mechanism of Time-Varying Viscosity Newtonian Fluid

### 3.1. Time-Varying Newtonian Fluid Penetration Equation of Motion

In a circular tube, with viscosity time-varying Newtonian fluid motion, assuming that the radius is $r_0$, a microfluidic column is taken in it, with the tube axis as the symmetry axis, the radius $r < r_0$, and the length is $dl$ [28]. Suppose that the pressures at the left and right ends of the $dl$ segment of the microfluidic column are $p$ and $p + dp$, respectively, so the pressure difference of the microflow section is $dp$; the shear stress on the surface is $\tau$, the direction of the shear stress is towards the left, and the flow velocity is in the opposite direction, as shown in Figure 1.

Under the gravity condition of ignoring the time-varying Newtonian fluid, the force of the microfluidic column shown in Figure 1 satisfies the following equilibrium relationship:

$$\pi r^2 dp + 2\pi r\tau dl = 0 \tag{4}$$

The shear stress $\tau$ on the surface of the microfluidic column:

$$\tau = -\frac{r}{2}\frac{dp}{dl} \tag{5}$$

Substituting Equation (5) into Equation (3) obtains:

$$\gamma = -\frac{dv}{dr} = \frac{\tau}{\eta_0 e^{kt}} = (-\frac{1}{2\eta_0 e^{kt}}\frac{dp}{dl})r \tag{6}$$

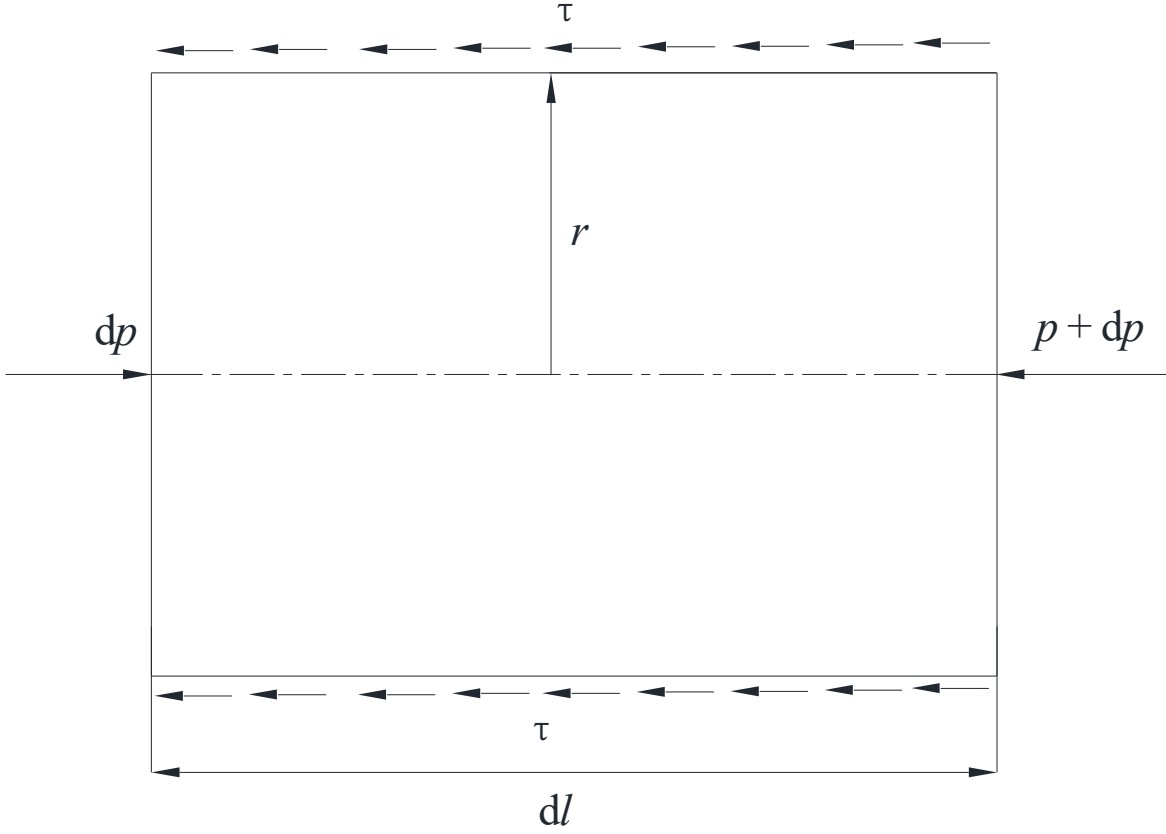

**Figure 1.** Flow diagram of a Newtonian fluid with time-dependent behavior in a circular tube.

Equation (6) is integrated by the method of separation of variables, and the boundary conditions $r = r_0$ and $v = 0$ at the pipe wall are considered. In this way, the velocity equation of the time-varying Newtonian fluid moving in the cross-section of the pipe is obtained as:

$$v = -\frac{1}{4\eta_0 e^{kt}}\frac{dp}{dl}(r_0{}^2 - r^2) \tag{7}$$

The flow rate, $Q_P$, for a time-varying Newtonian fluid with laminar motion in a circular tube, can be obtained as:

$$Q_P = \int_0^{r_0} 2\pi r v \, dr \tag{8}$$

Substituting Equation (7) into Equation (8) can obtain:

$$Q_P = -\frac{\pi r_0{}^4}{8\eta_0 e^{kt}}\frac{dp}{dl} \tag{9}$$

Therefore, the average velocity of the viscosity time-varying Newtonian fluid along the cross-section of the pipe is:

$$\overline{v} = \frac{Q_P}{\pi r_0^2} = -\frac{r_0{}^2}{8\eta_0 e^{kt}}\frac{dp}{dl} \tag{10}$$

Because the average penetration velocity of the actual movement of the fluid in the injected medium satisfies equation $V = \phi \overline{v}$ [28], and:

$$k_s = \frac{\phi r_0^2}{8} \tag{11}$$

where $k_s$ is the permeability of the injected medium, and $\phi$ is the porosity of the injected medium.

In geotechnical tests, the permeability coefficient, $K$, of porous media is usually used to characterize its permeability. The relationship between the permeability coefficient $K$, and permeability $k_s$, satisfies the following relationship [29]:

$$K = \frac{k_s \rho_\text{w} g}{\mu_\text{w}} \tag{12}$$

where $\mu_\text{w}$ is the viscosity of water, $\rho_\text{w}$ is the density of water and equals 1000 kg/m$^3$, $g$ is the gravitational acceleration and equals 9.8 m/s$^2$, and other symbols are as above.

The penetration motion equation of viscosity time-varying Newtonian fluid can be obtained by combining Equations (10–12):

$$V = e^{-kt} \left( \frac{K\mu_\text{w}}{\rho_\text{w} g \eta_0} \right) \left( -\frac{dp}{dl} \right) \tag{13}$$

Also ordered:

$$\beta = \frac{\eta_0}{\mu_\text{w}} \tag{14}$$

where $\beta$ is the ratio of slurry viscosity to water viscosity.

Thus, the penetration motion equation of viscosity time-varying Newtonian fluid can be transformed into:

$$V = \frac{e^{-kt}}{\rho_\text{w} g} \left( \frac{K}{\beta} \right) \left( -\frac{dp}{dl} \right) \tag{15}$$

where $V$ is the time-varying Newtonian fluid in the medium being injected with the actual movement of the average percolation velocity, and the other symbols are as above.

### 3.2. Columnar Penetration Grouting Mechanism of Time-Varying Viscosity Newtonian Fluid

According to the literature [30,31], the following assumptions are used in this paper to study the mechanism of columnar penetration grouting of a time-varying viscosity Newtonian fluid:

(1)  The medium or material being injected satisfies isotropy and homogeneity;
(2)  The fluid is incompressible and the flow pattern remains unchanged during the grouting process;
(3)  The dynamic viscosity of Newtonian fluids is time-varying (increasing with time), and it changes exponentially with time;
(4)  The grouting is carried out by the filling pressure method, and the slurry is injected into the injected medium from the side hole of the grouting pipe and diffuses in a columnar surface;
(5)  The gravity effect of the fluid is neglected in the grouting process, and the influence of the diffusion path of the slurry in the injected medium is not considered;
(6)  The flow velocity is small, and the slurry is laminar, except for the turbulent flow state in the local area around the grouting hole.

The diffusion theory model used to study the mechanism of columnar penetration grouting of a viscosity time-varying Newtonian fluid, is shown in Figure 2.

In Figure 2, $p_1$ is the grouting pressure, $p_0$ is the groundwater pressure at the grouting point, $l_1$ is the diffusion radius of the time-varying viscosity Newtonian fluid at time $t$, $l_0$ is the radius of grouting hole, and $m$ is the columnar diffusion height.

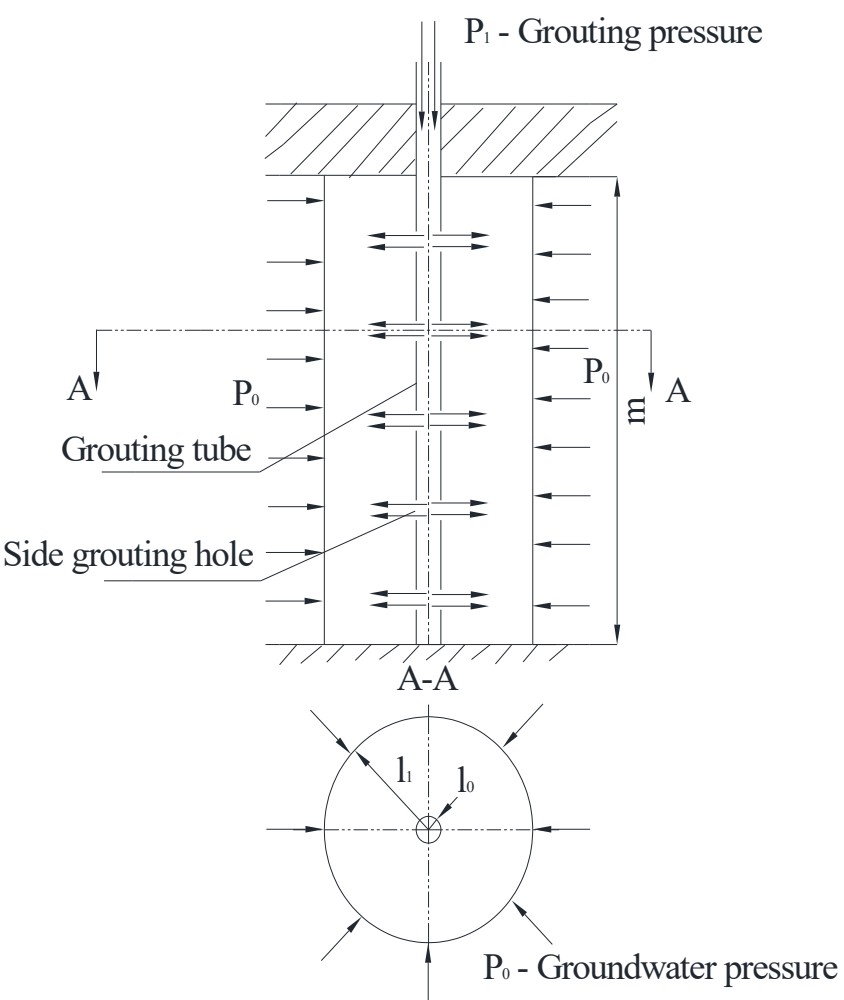

**Figure 2.** Column spreading model of a Newtonian fluid with time-dependent behavior of viscosity.

The total grouting amount, $Q$, of viscosity time-varying Newtonian fluids in the grouting process satisfies the following equation:

$$Q = VAt \tag{16}$$

where $A$ is the total surface area of the time-varying Newtonian fluid diffusion in the grouting area.

The total diffusion surface area, $A$, of the diffusion theoretical model shown in Figure 2 is:

$$A = 2\pi lm \tag{17}$$

Substituting Equations (16) and (17) into Equation (15), obtains:

$$dp = -\frac{\rho_w g e^{kt} Q \beta}{2\pi t lmK} dl \tag{18}$$

For Equation (18), the separation variable method is used to integrate and consider the grouting boundary conditions: when $p = p_1$, $l = l_0$; and $p = p_0$, $l = l_1$, then:

$$\Delta p = p_1 - p_0 = \frac{\rho_w g e^{kt} Q \beta}{2\pi t mK} \ln\left(\frac{l_1}{l_0}\right) \tag{19}$$

Because of the total amount of grouting $Q = \pi l_1^2 m\phi$, the TVNCPGDM equation can be obtained as:

$$\Delta p = p_1 - p_0 = \frac{\rho_w g e^{kt} \beta \phi l_1^2}{2tK} \ln \frac{l_1}{l_0} \tag{20}$$

where $p_1$ is grouting pressure (Pa), $p_0$ is the groundwater pressure at the grouting point (Pa), $t$ is the grouting time (s), $\phi$ and $K$ are the porosity (dimensionless number) and permeability coefficient (m/s) of the injected medium, respectively, $\rho_w$ is the density of water (kg/m$^3$), generally 1000 kg/m$^3$, $g$ is the acceleration of gravity (m/s$^2$), generally 9.8 m/s$^2$, $l_1$ is the diffusion radius (m) of the viscosity time-varying Newtonian fluid in the injected medium, and $l_0$ is the radius of the grouting hole (m).

Without considering the time-varying of the Newtonian fluid, and dividing $\rho_w g$ on both sides at the same time, Equation (20) can be simplified to obtain the current Newtonian fluid columnar penetration grouting, Equation [7]:

$$\Delta h = \frac{\beta \phi l_1^2}{2tK} \ln \frac{l_1}{l_0} \tag{21}$$

## 4. The Applicable Scope and Parameter Determination of the Equation

### 4.1. Application Scope of Equation

Equation (20) is derived based on the laminar motion state of Newtonian fluids, so it is not applicable to turbulent flow.

According to the literature [32–34], the laminar or turbulent flow state of viscosity time-varying Newtonian fluids, is determined using the Reynolds number, Re. When Re < 2000, the Newtonian fluid exhibits laminar flow; when Re > 4000, the Newtonian fluid is turbulent; when 2000 < Re < 4000, the Newtonian fluid is a mixed flow state of turbulent flow and laminar flow.

The Reynolds number, Re, can be calculated by Equation (22):

$$\mathrm{Re} = \frac{\overline{v}d}{\eta} \tag{22}$$

where $\overline{v}$ is the average velocity of the viscosity time-varying Newtonian fluid in the injected medium or material, $d$ is the size of the spatial range of its flow (such as: pipe radius or diameter, this article refers to the pore size of the time-varying Newtonian fluid flowing in the injected medium or material), the meaning of $\eta$ is shown above.

### 4.2. Determining the Parameters in the Equation

The determination method of each parameter to solve Equation (20) is as follows:

The permeability coefficient $K$, of the injected medium or material, reflects its permeability characteristics, which can be determined by indoor or field measurement methods. However, in order to truly reflect the permeability of the injected medium or material, it is often obtained by the field water injection test.

The grouting time $t$, can be designed and selected according to the actual engineering situation or field experience, and the grouting pipe radius $l_0$, can be determined by directly measuring it multiple times and taking the average value.

Porosity $\phi$ is the ratio of the pore volume to total volume in the injected medium or material, which can be obtained by the following equation:

$$\phi = \frac{\gamma_s(1+\omega) - \gamma}{\gamma_s(1+\omega)} \tag{23}$$

where $\omega$ is the water content of the injected medium or material, $\gamma$ is the natural weight, and $\gamma_s$ is the soil bulk density. They can be determined according to reference [29].

The initial dynamic viscosity $\eta_0$ and time-varying coefficient $k$, of the time-varying Newtonian fluid, can be obtained by one of the following two methods:

(1) Obtained based on currently available research results on the time-varying viscosity of Newtonian fluids, such as [7,35–37], etc.

(2) Measurement tests of Newtonian fluid viscosity are carried out using rotary or capillary viscometers and are obtained by analysis and calculation.

On the basis of obtaining the porosity $\phi$ and permeability coefficient $K$, of the injected medium or material, the $\mu_w$ parameter value is calculated by Equation (14) by querying the viscosity $\beta$ value of water at different temperatures.

At this point, the parameters needed in Equation (20) are completely determined. Under the condition of knowing the grouting pressure and grouting point groundwater pressure difference $\Delta p$, the theoretical diffusion radius, $l_1$, of columnar penetration grouting of the viscosity time-varying Newtonian fluid in the injected medium or material can be obtained. On the contrary, given $l_1$, the theoretical grouting pressure difference, $\Delta p$, can be obtained.

## 5. Grouting Verification Test

In order to analyze the accuracy and applicability of the TVNCPGDM equation (Equation (20)), the grouting experiment was designed to verify it.

### 5.1. Experimental Device and Grouting Material

The device and related instructions used in the grouting test, refer to the previous research results of the team in [38].

In this paper, the grouting test apparatus, consisting of a pressure supply unit, slurry storage unit, and experimental box, is used to carry out the grouting test on gravel soil. A schematic diagram of the grouting test device is shown in Figure 3.

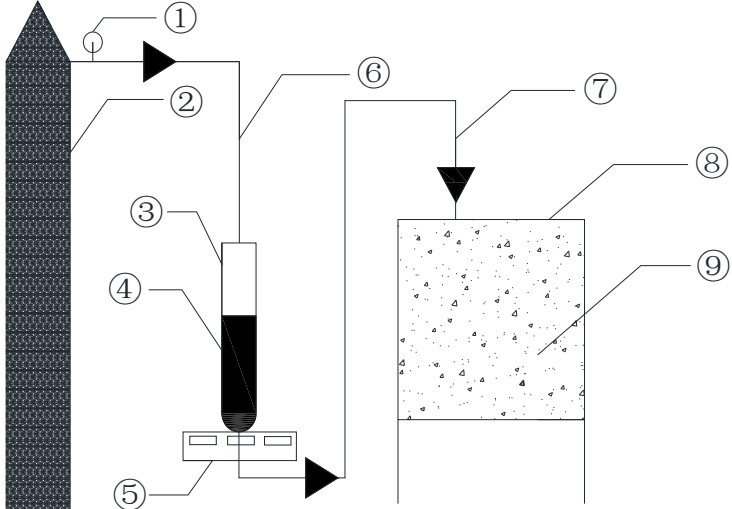

**Figure 3.** Schematic diagram of the grouting verification test device. ①—Nitrogen pressure reducer and grouting control switch; ②—pressure supply equipment; ③—equipment for storing grouting slurry; ④—Newtonian fluid slurry; ⑤—electronic scale; ⑥—grouting duct; ⑦—grouting pipe; ⑧—experimental box; ⑨—the sand gravel.

The pressure supply system uses a pressurized nitrogen tank to provide the pressure for grouting, and a pressure reducer on the nitrogen tank to control and monitor the pressure. The slurry storage equipment is used to store the time-varying Newtonian fluid slurry. The electronic scale in the lower part can monitor the quality of the time-varying Newtonian fluid injected into the experimental box during the whole grouting test, in real time. A photograph of the grouting test apparatus is shown in Figure 4.

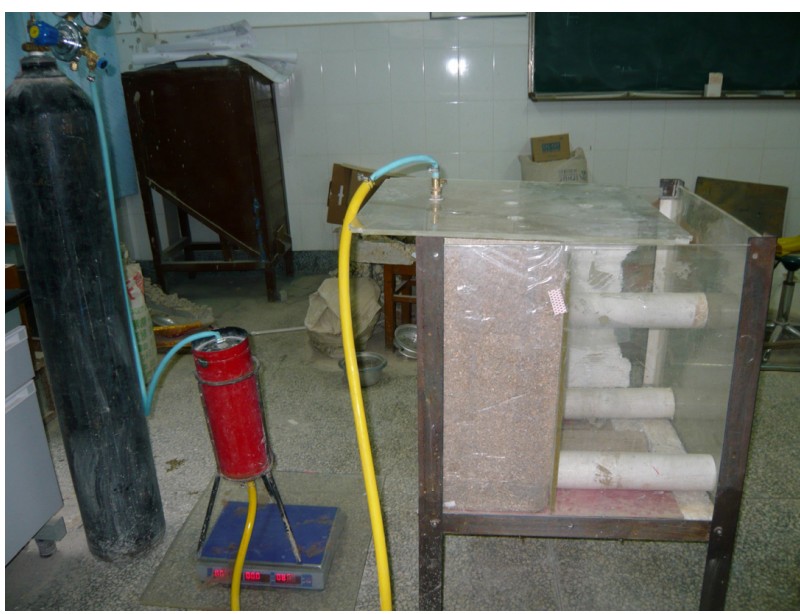

**Figure 4.** Photograph of the grouting experimental device.

The grouting material is #32.5 ordinary Portland cement, produced by Kunming Cement Plant. Cement is widely used as the grouting material in current engineering. In the verification experiment of this paper, the cement slurry, with a water–cement ratio of 2.0, is configured to carry out the grouting verification test. According to the literature [34,35], this cement slurry is a typical Newtonian fluid; the rheological equation and viscosity time-varying equation come from the relevant research results in references [36,39].

In this paper, sand gravel with a particle size distribution in the range of 3–5 mm is selected as the injection medium for the verification experiment. The specific gravity of the injected sand gravel is 2.65, the mass water content is 2.79%, the density is 1496.80 kg/m$^3$, and the porosity and permeability coefficient are 45.05% and $2.11 \times 10^{-2}$ m/s, respectively. In order to make the injected sand gravel meet the isotropic and homogeneous assumptions to the maximum extent, it was washed three times in clear water before the grouting test.

*5.2. Grouting Verification Test*

5.2.1. Experimental Design

The indoor environment temperature of the test, and the water temperature of the cement slurry, are close to 25 °C (at 25 °C, the viscosity of water is $0.89 \times 10^{-3}$ Pa · s). The designed grouting pressure is 1500 Pa, the groundwater pressure in the sand gravel is 0 Pa, and the grouting time is 270.5 s. The grouting pipes used in the grouting test are PVC pipes with a diameter of 15 mm, blocking the bottom grouting holes. The number of side grouting pipes is three, and the distribution is shown in Figure 5.

5.2.2. Determination of Grouting Diffusion Radius

A photo of the grouting verification test is shown in Figure 6.

After completing the grouting test with the designed parameters, the grouted stone body formed, consolidated, and completely dried; then, the test box is disassembled to measure the actual diffusion radius value, with a ruler or tape. Significantly, the standard deviation is controlled within 5% for at least three measurements, and then their average values are calculated.

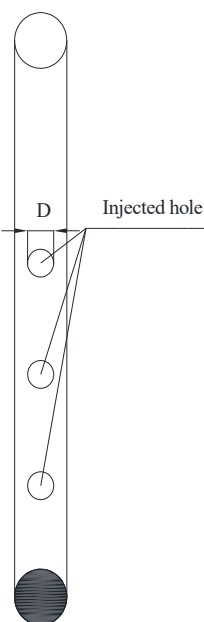

**Figure 5.** Grout pore distribution diagram of grouting pipe.

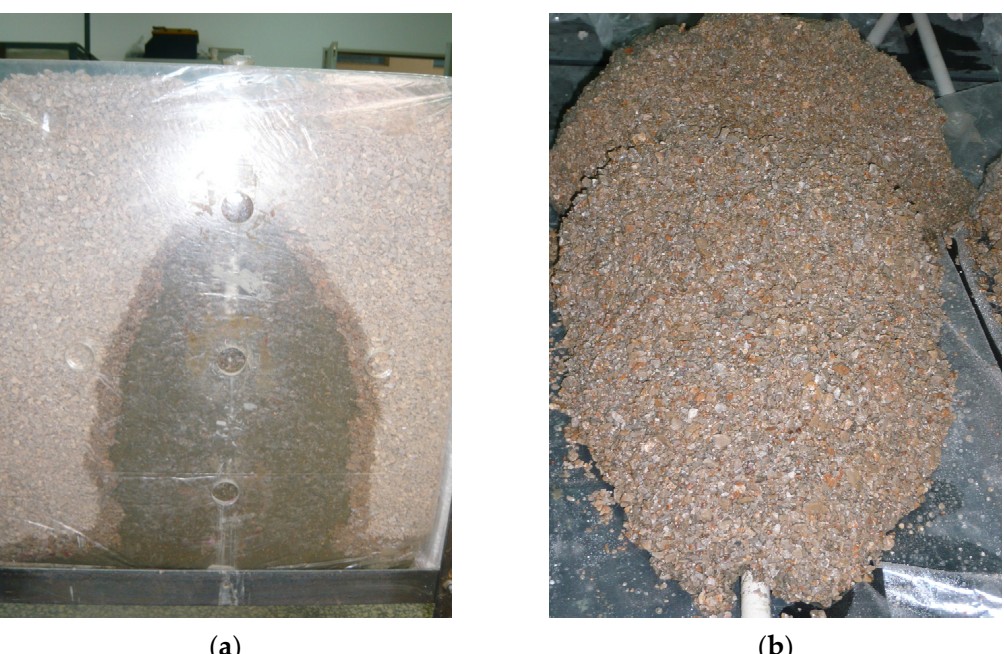

(**a**)                                                              (**b**)

**Figure 6.** Photographs of grouting experimental process. (**a**) Permeation and diffusion form of Newtonian cement slurry in the gravel sand; (**b**) the grouted stone body.

The grouting diffusion radius is determined according to the method introduced in reference [40].

### 5.2.3. Analysis of Results

The theoretical calculation value and the actual measurement value of the diffusion radius are shown in Table 1, with the theoretical calculation value obtained according to the TVNCPGDM equation. For comparison, the theoretical value of the diffusion radius calculated by the columnar penetration grouting equation (Equation (7)), without considering the time-varying viscosity of Newtonian fluids, is also listed in Table 1.

**Table 1.** Theoretical calculation value and actual measurement value of diffusion radius of the grouting test.

| According to the Equation of this Paper, the Calculated Value of Grouting Diffusion Radius Theory /mm | According to the Calculated Value of the Columnar Penetration Grouting Equation without Considering the Time-Varying Viscosity of Newtonian Fluid in Reference [7] /mm | Grouting Diffusion Radius Actual Measured Value /mm |
|---|---|---|
| 245.3 | 261.6 | 126.3 |

The analysis of Table 1, shows that the theoretical value of the diffusion radius calculated by the TVNCPGDM equation derived in this paper, is closer to the experimental value than the theoretical value of the diffusion radius obtained by the columnar penetration grouting equation without considering the time-varying viscosity of the Newtonian fluid. This shows that the TVNCPGDM equation derived in this paper can better reflect the columnar penetration grouting diffusion law and diffusion pattern of Newtonian fluids in the injected medium with time. In addition, the obtained penetration grouting diffusion radius is more in line with the actual grouting engineering requirements, so it can provide theoretical support and guidance for the actual grouting construction.

There is still a large gap between the theoretical diffusion radius of grouting calculated by the TVNCPGDM equation and the grouting test value. This is mainly due to the following four aspects [37,38]:

(1) In this paper, sand gravel is selected as the injection medium. Although their particle size distribution is uniform and they are washed three times before the test to meet the isotropy and homogeneity conditions as much as possible, they are still different from the isotropy and homogeneity assumptions made in theoretical research.

(2) Many factors affect the permeability and diffusion effect of viscosity time-varying Newtonian fluids in the injected medium. For example, when the cement slurry is configured, it is an unstable slurry, due to the excessive water separation rate, and it is assumed to be a stable slurry when the columnar permeability and diffusion mechanism equation of a viscosity time-varying Newtonian fluid is used to calculate the theoretical value. For example, problems such as precipitation and blockage may occur when cement slurry diffuses in the injected medium.

(3) At present, although some research results have been obtained on the time-varying characteristics of viscosity of Newtonian fluid, they cannot fully reflect the time-varying law. In the future, we should further strengthen the research on the time-varying characteristics and laws of Newtonian fluids.

(4) The diffusion path of Newtonian fluids in the injected medium, and the influence of gravity on the grouting effect, is not considered, which is the most important reason.

## 6. Conclusions

(1) Based on the basic rheological equation of Newtonian fluids and its dynamic viscosity time-varying law, the penetration motion equation of a viscosity time-varying Newtonian fluid is studied.

(2) Based on the penetration motion equation of a viscosity time-varying Newtonian fluid, the TVNCPGDM equation is derived, the application scope of the equation is analyzed, and the specific acquisition method of the parameters is determined.

(3) A grouting experiment was designed, to verify the derived TVNCPGDM equation. The results show that the theoretical value of the grouting diffusion radius calculated by the TVNCPGDM equation is closer to the experimental value than the theoretical value of the diffusion radius obtained by the columnar penetration grouting equation without considering the time-varying viscosity of the Newtonian fluid. This shows that the TVNCPGDM equation derived in this paper can better reflect the diffusion law and diffusion morphology of columnar penetration grouting of Newtonian fluids

in the injected medium with time, and the obtained diffusion radius of penetration grouting is more in line with the actual grouting engineering needs, so it can provide theoretical support and guidance for actual grouting construction.

**Author Contributions:** Conceptualization, Z.Y. and S.W.; methodology, X.Z. and R.L.; validation, X.M., H.X. and W.X.; formal analysis, H.X. and W.X.; investigation, R.L. and T.X.; resources, S.W., C.Y. and J.Y.; data curation, X.M., R.L. and J.C.; writing—original draft preparation, Z.Y.; writing—review and editing, X.Z. and X.W.; visualization, X.Z., C.Y. and J.Y.; supervision, P.W., X.L. and J.C.; project administration, X.L. and T.X. All authors have read and agreed to the published version of the manuscript.

**Funding:** This research was funded by the National Natural Science Foundation of China (Grant No. 41861134008), the Muhammad Asif Khan academician workstation of Yunnan Province (Grant No. 202105AF150076), the Key R&D Program of Yunnan Province (Grant No. 202003AC100002), the General Program of basic research plan of Yunnan Province (Grant No. 202001AT070043), the Science and Technology Talents and Platform Program of Yunnan Province (Grant No. 202305AD160064), the Basic Research Project of Yunnan Province (Grant No. 202201AT070283), Kunming University of Science and Technology Top Innovative Talents Project, and Analysis and Testing Fund of Kunming University of Science and Technology (Grant No. 2022M20212239003).

**Data Availability Statement:** Data are contained within the article.

**Conflicts of Interest:** The authors declare no conflict of interest.

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
