# Peer review of "Study on Mechanism and Verification of Columnar Penetration Grouting of Time-Varying Newtonian Fluids"

_processes, doi:10.3390/pr11041151_

Round 1
Reviewer 1 Report
In the abstract, it should be explained why this study is needed, what kind of gap it will fill in the literature or industry, and whether this result has been achieved by giving some of the results.
Please explain how the equation (7) is obtained.
Please give name of the parameters in equations 13, 14, 15.
In the article, mainly theoretical information is given. The results obtained from the experiments have not been adequately examined. What kind of theoretical or industrial contribution these results have made is not discussed. Whether the diffusion radius obtained from the experiments is close to the theoretical results should be examined by numerical analysis methods.
The results of the experiment to find the parameters of the formula used to reach the diffusion radius in Table 1 and these values should be given.
Experimental conditions are not clear enough. The number of times the experiment was repeated is unclear. Has a statistical analysis been done on these results?
Author Response
Dear editors and reviewers:
Thank you for your and reviewers’ comments concerning our manuscript entitled “Study on mechanism and verification of columnar penetration grouting of time-varying Newtonian fluid” (ID: 2249109).

Reviewer 2 Report
Dear Authors
Despite my sincere intentions and respect for the authors' work, I am unfortunately forced to reject this article. I have noticed multiple editorial errors. The layout does not comply with the instructions, the division into sections is incorrect. Citations in the text are also not correct. I observed different font sizes in the text. However, these comments are to be corrected quickly. I am not a mathematician so I cannot comment on the derivation of the formula for modelling time-dependent Newtonian fluids. My serious reservations are with the verification part of the experiment with cement slurry to penetrate the fine aggregate. By the way, this process is just called penetration and not diffusion. Getting back to the point, I do not really understand how the validity of a mathematical reality can be based on a single experiment. To make their calculations credible, a whole series of experiments should have been planned and carried out, depending on critical factors such as the type of cement, amount of water, aggregate grain size curve, temperature, pressure, etc. I also don't really understand why the authors call the material they tested bio-cement. In summary, the article contains critical experimental errors and should be supplemented by a series of reliable studies. I cannot recommend this article for publication.
Sincerely
Reviewer
Author Response
Manuscript 2249109 Response to Editors and Reviewers
Dear editors and reviewers:
Thank you for your and reviewers’ comments concerning our manuscript entitled “Study on mechanism and verification of columnar penetration grouting of time-varying Newtonian fluid” (ID: 2249109) Those comments are all valuable and so helpful for revising and improving our paper. We have studied carefully the comments and made corresponding answers and corrections which we expect to meet the approval. The revised contents are marked in red in the ‘Revised Manuscript with Changes Marked.docx’. Also, the point-by-point explanations to the comments, suggestions and questions are listed as below.
Dear Reviewer
We have improved the manuscript according to your valuable opinions. We have improved the editing errors in the manuscript, modified the layout, enhanced the relevance of the chapters in the manuscript, and modified parts of the manuscript. And there is a lot to explain to you. Firstly, the inconsistency of the font size in the manuscript is due to the problem of the Mathtype formula editor, which makes the font size inconsistent. Secondly, the experiments performed in our manuscript are in reference to the team's previous research results, and not just one experiment in the manuscript. In the previous study, we performed a rich experimental verification, detailed our experimental conditions, and theoretical analysis of the results. (Guo, T.; Zhang, Z.; Yang, Z.; Zhu, Y.; Yang, Y.; Guo, Y.; Wang, R.; Zhang, B.; Fang, Y.; Yu, D. et al. Penetration Grouting Mechanism of Time‐Dependent Power‐Law Fluid for Reinforcing Loose Gravel Soil. Minerals 2021, 11, 1391.). Thirdly, the TVNCPGDM equation derived in this paper is closer to the experimental results than According to the calculated value of the columnar grouting equation without considering the time-varying viscosity of Newtonian fluid in Reference. Fourth, our experiment is indeed what you call a penetration grouting test, not a diffusion grouting test, and the part that is not clearly described has been corrected in the manuscript. Fifth, the experimental material we used in the experiment is 32.5 ordinary Portland cement produced by Kunming Cement Plant, not the biological cement you mentioned (Chapter 5.1). Sixth, the type of cement, temperature, pressure and so on are described in detail in the previous experiments of our team, and also described in detail in Chapter 5.1 of this paper. All in all, we have made a comprehensive revision of the manuscript according to your opinion. We sincerely hope this manuscript will be finally acceptable to be published on MDPI.
Sincerely
Zhao Xuguang

Reviewer 3 Report
Reviewer Comments
Authors have contributed well the transient non-Newtonian fluid analysis with title “Study on mechanism and verification of columnar penetration grouting of time-varying Newtonian fluid”. Some valuable comments are given below;
- The authors should cite at least one concrete potential application where the present results are likely to be relevant, howsoever remotely, over this range of conditions.
- Precise why the study is unsteady and non-Newtonian?
- All the assumptions should be explicitly mentioned with proper justification.
- The introduction needs to enrich the readers with state-of-the-art works, which gives the reader a clear vision of the gap that those studies have not addressed and covered by this study.
- Extensive validation of the solver needs to be included in the manuscript.
- What is about a domain independence study? I suggest the authors should add a domain independence study.
- In the last paragraph of the introduction, the innovation of the present work should be stated precisely. In my opinion, this section should be completely rewritten.
Improve the abstract section by using the core and quantitative results.

Author Response
Manuscript 2249109 Response to Editors and Reviewers
Dear editors and reviewers:
Thank you for your and reviewers’ comments concerning our manuscript entitled “Study on mechanism and verification of columnar penetration grouting of time-varying Newtonian fluid” (ID: 2249109) Those comments are all valuable and so helpful for revising and improving our paper. We have studied carefully the comments and made corresponding answers and corrections which we expect to meet the approval. The revised contents are marked in red in the ‘Revised Manuscript with Changes Marked.doc’. Also, the point-by-point explanations to the comments, suggestions and questions are listed as below.
Q1. The authors should cite at least one concrete potential application where the present results are likely to be relevant, howsoever remotely, over this range of conditions.
Answer: This comment was supplemented according to the reviewer's suggestions, as detailed in introduction of the revised manuscript.
Q2. Precise why the study is unsteady and non-Newtonian?
Answer: Thank you for your professional suggestions. First of all, what we study in the manuscript is time-varying refers to the change over time, specifically refers to the cement slurry will change over time can be predicted, not that the cement slurry is unstable. Secondly, we study Newtonian fluid in the manuscript, not non-Newtonian fluid.
Q3. All the assumptions should be explicitly mentioned with proper justification.
Answer: Thank you so much for your suggestions. The assumptions mentioned in the manuscript have been clearly expressed and have a clear literature basis. And many references and books are described in this way. If the whole content of the literature is added to the manuscript, the readability of the manuscript will become worse and the main content cannot be displayed. ([1] Shen, C.-T.; Liu, H.-N. Non-Newtonian fluid mechanics and its application; Higher Education Press: Beijing, China, 1989. [2] Gao, D.-R. Engineering Fluid Mechanics; Mechanical Industry Press: Beijing, China, 1999. [3] Guo, T.; Zhang, Z.; Yang, Z.; Zhu, Y.; Yang, Y.; Guo, Y.; Wang, R.; Zhang, B.; Fang, Y.; Yu, D. et al. Penetration Grouting Mechanism of Time‐Dependent Power‐Law Fluid for Reinforcing Loose Gravel Soil. Minerals 2021, 11, 1391. [4] Lu, Q.; Yang, Z.-C.; Yang, Z.-C.; Yu, R.-X.; Zhu, Y.-Y.; Yang, Y.; Zhang, B.-H.; Wang, R.-C.; Fang, Y.-C.; Yu, D.-L.; Liu, H.; Su, J.-K. Penetration grouting mechanism of Binham fluid considering diffusion paths. Rock Soil Mech. 2022, 43(02), 385-394.)
Q4. The introduction needs to enrich the readers with state-of-the-art works, which gives the reader a clear vision of the gap that those studies have not addressed and covered by this study.
Answer: Thank you very much for your comments. We have revised the introduction of the manuscript.
Q5. Extensive validation of the solver needs to be included in the manuscript.
Answer: We appreciate your professional advice on our paper. The experiments we describe in the manuscript were not concluded by conducting only one experiment, but were verified by referring to the team's previous research results. In the team's previous research, we conducted detailed experiments and performed statistical analysis of the results. (Guo, T.; Zhang, Z.; Yang, Z.; Zhu, Y.; Yang, Y.; Guo, Y.; Wang, R.; Zhang, B.; Fang, Y.; Yu, D. et al. Penetration Grouting Mechanism of Time‐Dependent Power‐Law Fluid for Reinforcing Loose Gravel Soil. Minerals 2021, 11, 1391.)
Q6. What is about a domain independence study? I suggest the authors should add a domain independence study.
Answer: We appreciate your suggestions. We have revised the introduction of the manuscript at the request of the reviewer.
Q7. In the last paragraph of the introduction, the innovation of the present work should be stated precisely. In my opinion, this section should be completely rewritten.
Answer: Thank you so much for your advice. We have revised the introduction of the manuscript at the request of the reviewer.
Q8. Improve the abstract section by using the core and quantitative results.
Answer: Thank you so much for your suggestion. We have modified the abstract according to the requirements of the reviewer.

Reviewer 4 Report
Recommendations:
1- write an application of Newtonian fluid.
2- Regarding the Mathematical Formulation sections: appropriate references should be presented for some equations.
3- I need clarification on the application of the problem studied.
4- The Introduction section should include a proper application regarding the present problem.
5- In the same section, the authors may include more discussions regarding mentioned previous works.
6- There are many grammar and typo mistakes throughout the manuscript. The authors should revise the entire paper carefully before considering it for publication.
7. The literature review can be updated considering recent studies
https://doi.org/10.1007/s13369-020-05195-x
https://doi.org/10.1007/s13369-021-06092-7
https://doi.org/10.1007/s13369-021-06412-x
https://doi.org/10.1140/epjp/s13360-020-00606-2
https://doi.org/10.1080/17455030.2022.2050441
https://doi.org/10.1088/1402-4896/ac03de
DOI: 10.1177/09544089211072715
DOI: 10.1177/09544089221105932
https://doi.org/10.1080/17455030.2022.2072536
DOI: 10.1177/09544089221139696
https://doi.org/10.1080/17455030.2022.2100004
Author Response
Manuscript 2249109 Response to Editors and Reviewers
Dear editors and reviewers:
Thank you for your and reviewers’ comments concerning our manuscript entitled “Study on mechanism and verification of columnar penetration grouting of time-varying Newtonian fluid” (ID: 2249109) Those comments are all valuable and so helpful for revising and improving our paper. We have studied carefully the comments and made corresponding answers and corrections which we expect to meet the approval. The revised contents are marked in red in the ‘Revised Manuscript with Changes Marked.doc’. Also, the point-by-point explanations to the comments, suggestions and questions are listed as below.
Q1. Write an application of Newtonian fluid.
Answer: This comment was supplemented according to the reviewer's suggestions, as detailed in Introduction of the revised manuscript.
Q2. Regarding the Mathematical Formulation sections: appropriate references should be presented for some equations.
Answer: Thank you for your professional suggestions. The source and derivation process of all Equations have been shown in the manuscript, and references have been marked. And we have modified the places where the description is unclear and the variable name is not given in the manuscript.
Q3. I need clarification on the application of the problem studied.
Answer: Thank you so much for your suggestions. We have revised the introduction of the manuscript at the request of the reviewer.
Q4. The Introduction section should include a proper application regarding the present problem.
Answer: Thank you very much for your comments. We have revised the introduction according to the requirements of the reviewer.
Q5. In the same section, the authors may include more discussions regarding mentioned previous works.
Answer: We appreciate your professional advice on our paper. We try to increase the discussion of the work done in this section in each section of the manuscript, but this will change the logic of the manuscript, make the focus of the article unclear, and cause readers to read difficultly.
Q6. There are many grammar and typo mistakes throughout the manuscript. The authors should revise the entire paper carefully before considering it for publication.
Answer: We appreciate your suggestions. We have modified the whole article according to the requirements of the reviewer.
Q7. The literature review can be updated considering recent studies.
Answer: Thank you so much for your advice. We have carefully read the references recommended by the reviewers and selected the most relevant ones to be cited in the introduction of the manuscript.

Round 2
Reviewer 1 Report
The test results of the study should be given and discussed in detail. It is necessary to make comments based on the literature by comparing your theoretical presentations with your test results.
Author Response
Manuscript 2249109 Response to Editors and Reviewers
Dear editors and reviewers:
Thank you for your and reviewers’ comments concerning our manuscript entitled “Study on mechanism and verification of columnar penetration grouting of time-varying Newtonian fluid” (ID: 2249109) Those comments are all valuable and so helpful for revising and improving our paper. We have studied carefully the comments and made corresponding answers and corrections which we expect to meet the approval. The revised contents are marked in red in the ‘Revised Manuscript with Changes Marked.docx’. Also, the point-by-point explanations to the comments, suggestions and questions are listed as below.
Dear Reviewer
We have improved the manuscript according to your valuable opinions. In Chapter 5.1, the theoretical diagram and the physical diagram of the experimental device are added to complement our experimental process. In Chapter 5.2.2, the determination of the diffusion radius of the grout is described in more detail to support our experimental conclusion. In Chapter 5.2.3 we use the table to intuitively analyse the calculation results we have obtained and prove that our results are closer to the experimental values than those calculated using the previous formulae(Kuang, J.-Z. Geotechnical grouting theory and engineering examples; Science press: Beijing, China, 1993.). The reason why the calculated results of the formula derived in this paper are far from the experimental values is also analyzed. All in all, we have made a comprehensive revision of the manuscript according to your opinion. We sincerely hope this manuscript will be finally acceptable to be published on MDPI.
Sincerely
Zhao Xuguang

Reviewer 2 Report
Dear Authors
Thank you for the clarification, however, the article still contains errors. There should be a corrected layout and descriptions on the drawings. You still use the phrase "bio-claystone" which is false (5.2.2). Further there is also used the concept of diffusion which you agreed to change. However, these are small changes.
I have no answer to the question of how many verification tests have been conducted. The values tested should be accompanied by measurement statistics. Then we can talk about the validation of the theoretical equations under near-real conditions. This is the key information for me.
Best regards
Author Response
Manuscript 2249109 Response to Editors and Reviewers
Dear editors and reviewers:
Thank you for your and reviewers’ comments concerning our manuscript entitled “Study on mechanism and verification of columnar penetration grouting of time-varying Newtonian fluid” (ID: 2249109) Those comments are all valuable and so helpful for revising and improving our paper. We have studied carefully the comments and made corresponding answers and corrections which we expect to meet the approval. The revised contents are marked in red in the ‘Revised Manuscript with Changes Marked.docx’. Also, the point-by-point explanations to the comments, suggestions and questions are listed as below.
Dear Reviewer
We have improved the manuscript according to your valuable opinions. In Chapter 5.1, the theoretical diagram and the physical diagram of the experimental device are added to complement our experimental process. In Chapter 5.2.2, the expression of ' bio-claystone ' is updated. Previously, our expression was not clear, and it was changed to ' grouted stone body '. In Chapter 5.2.2, the determination of the diffusion radius of the grout is described in more detail to support our experimental conclusion. In Chapter 5.2.3 we use the table to intuitively analyse the calculation results we have obtained and prove that our results are closer to the experimental values than those calculated using the previous formulae(Kuang, J.-Z. Geotechnical grouting theory and engineering examples; Science press: Beijing, China, 1993.). The reason why the calculated results of the formula derived in this paper are far from the experimental values is also analyzed. All in all, we have made a comprehensive revision of the manuscript according to your opinion. We sincerely hope this manuscript will be finally acceptable to be published on MDPI.
Sincerely
Zhao Xuguang

Round 3
Reviewer 1 Report
But experiment input data and results are not given and they are not discussed. In addition, the design of the article does not comply with international norms, it needs to be improved.
Reviewer 2 Report
After taking into account the corrections, I recommend the article for publication